# Prenatal Cocaine Exposure, Perinatal Risks, and Mediators to Preadolescent Attention Deficit Hyperactivity Disorder (ADHD)

**DOI:** 10.3390/children12111570

**Published:** 2025-11-19

**Authors:** Thitinart Sithisarn, Carla M. Bann, Barry Lester, Seetha Shankaran, Toni Whitaker, Rosemary D. Higgins, Henrietta Bada

**Affiliations:** 1Department of Pediatrics, University of Kentucky, Lexington, KY 40508, USA; hbada2@uky.edu; 2Statistics and Epidemiology, Research Triangle Institute, RTI International, Durham, NC 27709, USA; cmb@rti.org; 3Department of Psychiatry and Human Behavior, Brown University, Providence, RI 02903, USA; barry_lester@brown.edu; 4Center for the Study of Children at Risk, Providence, RI 02903, USA; 5Department of Pediatrics, School of Medicine, Wayne State University, Detroit, MI 48201, USA; sshankar@med.wayne.edu; 6Department of Pediatrics, University of Tennessee Health Science Center, Memphis, TN 38163, USA; 7Office of Research, Florida Gulf Coast University, Fort Myers, FL 33965, USA; rhiggins@fgcu.edu

**Keywords:** prenatal cocaine exposure, ADHD, behavioral problems, attention problem, impulsivity, path analysis

## Abstract

**Highlights:**

**What are the main findings?**

**What is the implication of the main finding?**

**Abstract:**

Attention-deficit/hyperactivity disorder (ADHD) is the most common behavioral problem in children. Multiple risk factors, including prenatal substance exposure, have been associated with this disorder. Objectives: We determined (1) the rate of ADHD in children with prenatal cocaine exposure (PCE) as compared to those non-exposed, (2) the association of ADHD with the infant’s sex, race, and birth weight, maternal age and education, and other known risk factors, and factors that may mediate the relationship between these risk factors and ADHD. Methods: This was a secondary analysis of data from the Maternal Lifestyle Study for a long-term follow-up. ADHD was defined as any diagnosis of attention deficit, hyperactivity disorder, or the combination, from the National Institute of Mental Health Diagnostic Interview Schedule for Children (NIMH DISC) administered to children, ages 11 or 14 years. The main exposure variable was PCE. Independent variables included infant and maternal characteristics, caretaker psychopathology, and maternal–child conflict. Mediators evaluated were the child’s impulsivity at 4 years of age and attention problems at 5 years from the Child Behavior Checklist. Results: Path analysis revealed that the effects of risk variables, including PCE, were mediated through the child’s attention problems at age 5 years. Child’s impulsivity, which was significantly associated with attention problems, was also a mediator between PCE and ADHD. Male sex had a direct path to ADHD. Conclusions: Our findings lend support to early screening before 4 years of age in children with PCE or other risk factors for ADHD. Behavioral interventions provided during early childhood may mitigate the later diagnosis or severity of ADHD.

## 1. Introduction

Attention-deficit/hyperactivity disorder (ADHD) is one of the most commonly diagnosed childhood neurodevelopmental disorders in primary care [1]. Approximately two-thirds of children with ADHD continue to experience symptoms into adulthood, contributing to a significant burden on the individual, families, and society, and frequently co-occurring with other psychiatric conditions [2]. According to the Global Burden of Disease Study, the global age-standardized incidence and prevalence of ADHD in 2019 were 0.061% and 1.13%, respectively, with 2.5 times higher rates observed in male children [3]. The incidence of ADHD peaked between ages 5 and 9 years, with a prevalence of 5.4% among children and adolescents [3]. In the United States, data from the National Survey of Children’s Health (2016–2019) showed a prevalence of ADHD diagnosis in children aged 13 to 17 of 9.8%, increasing to 11.3% in 2022–2023 [4]. In addition, over half of these children (53%) are receiving medications. Fifty-nine percent with current ADHD diagnosis are classified as having the moderate to severe forms [5], of whom 77.9% estimated to have at least one co-occurring disorder [6,7]; two-thirds of children have persistent functional impairment or a subthreshold of impairing symptoms [8,9] into adulthood. Also, ADHD contributes to the likelihood of chronic health issues such as obesity, eating disorders, substance use, and risky sexual behavior [10].

Numerous factors may affect the reported prevalence variation in ADHD. Maternal and perinatal risk factors such as maternal age and education, and infant’s sex, race, low birthweight, and prematurity [11] are associated with ADHD [9,12]. There is an increasing interest in the role of genetics and epigenetics in the development and trajectory, diagnosis, and treatment of ADHD [6,13]. Perinatal substance exposure has also been associated with a higher risk of ADHD in childhood [14].

We previously reported on behavior problems associated with prenatal cocaine exposure (PCE). Serial assessments of behavior problems using the parent or caretaker response to the Child Behavior Checklist (CBCL) showed that PCE had a sustained effect on broadband measures, externalizing, internalizing, and total behavior problems through 7 years of age from the Maternal Lifestyle Study (MLS) [15]. Additionally, administration of CBCL to both the child’s teacher and parent or caregiver yielded elevated attention problem scores in PCE, although an ADHD diagnosis was not established in the cohort [16]. In using the screening tool, the Pediatric Symptom Checklist [17,18], we found that high PCE was associated with attention problems as early as 8 years of age [19]. In a subset of 234 children, 10 years of age, who were retrospectively selected from the MLS cohort for having newborn withdrawal manifestations from prenatal substance exposure, Miller and Anderson reported that 15% of these children had ADHD; the information was derived from the children’s interval medical history [20]. The study was descriptive and no analysis was performed to assess the odds of ADHD in the presence of specific risk factors including PCE.

The MLS follow-up extended to 15 years. By a prospective study design, the outcome ADHD was specifically assessed in children at ages 11 and 14 years using a computer-assisted diagnostic interview [21]. Therefore, for this study we determined (1) whether children with PCE have higher rates of ADHD than non-exposed children to cocaine or other drugs, (2) any significant association between sex, race, maternal age, and education, and other risk factors and ADHD, and (3) whether the child impulsivity and inattention would mediate the relationship between PCE and a subsequent diagnosis of ADHD during adolescence. Identification of risk factors and mediating conditions may facilitate early intervention and contribute to improved long-term outcomes.

## 2. Materials and Methods

The study is a secondary analysis of data from the MLS, a multicenter longitudinal study of 1388 enrolled children with prenatal cocaine and/or opiate exposure and their comparison cohort through age 15 years [22,23]. The study had approval from the Institutional Review Board of each of the four participating centers and the two coordinating centers (Research Triangle Institute and Brown University). A parent or legal guardian gave consent for the study, and children ≥ 7 years gave assent. Each site screened for prenatal cocaine and opiate exposure in study participants from 1993 to 1995 (*n* = 11,811), using meconium testing and maternal interview by the trained interviewers; the details of the interview process have been previously published. Of enrolled participants screened for exposure, 1388 mothers gave consent for enrollment in the long-term follow-up phase, which began when the child was 1 month of age, corrected for preterm birth. Exposed children were those whose mothers admitted to any cocaine or opiate use during pregnancy or whose meconium showed cocaine or opiate metabolites.

Comparison children were chosen within each site and matched for gestational age, sex, race, and ethnicity. Details on the MLS procedures have been published [22]. Children were followed at periodic intervals from enrollment at 1 month to 15 years of age.

For this study, the major outcome of interest was ADHD. We defined ADHD as any diagnosis of attention deficit or hyperactivity disorder, or combination, from the National Institute of Mental Health Diagnostic Interview Schedule for Children (NIMH DISC) [21], administered to children at ages 11 or 14 years. The main exposure variables included levels of PCE (high and some) with or without other drug exposures [22,24]. High PCE referred to maternal use ≥ 3 times per week in the first trimester, and some cocaine use referred to any other use. Because most women in the study who used cocaine during pregnancy also used marijuana, opiates, alcohol, tobacco, or other drugs, we created a prenatal drug exposure variable with four categories [23]: (1) high PCE with other drug exposure (high PCE/OD), (2) some PCE/OD, (3) no PCE but with exposure to OD (PCE−/OD+), and (4) no PCE and no OD exposure (PCE−/OD−) [23]. The total number of participants was 853: 95 (11.1%) in high PCE/OD, 198 (23.2%) in some PCE/OD, 343 (40.2%) in PCE−/OD+, and 217 (25.4%) in PCE−/OD− groups, respectively (Table 1).

From the many assessments during each follow-up visit, we selected the following child and maternal independent variables: child’s sex, ethnicity, birth weight < 2500 or ≥2500 g, maternal age at delivery, maternal education, caretaker psychopathology, and caretaker–child conflict. Caretaker psychopathology was determined from the Brief Symptom Inventory (BSI) [25], a questionnaire administered by interview to the caregiver to reflect the psychological symptom patterns of psychiatric and medical patients as well as community non-patient respondents. Caretaker–child conflict refers to the conflict summary scale derived from the questionnaire My Child’s Relationship With Me, administered to assess a parent’s perception of their relationship with their child [26,27]. Other variables included in the analysis were possible mediators, the child’s impulsivity at 4 years and attention problems at 5 years. The child’s impulsivity score was derived from the Child Behavior Questionnaire (CBQ), a questionnaire administered to the caregiver to measure the child’s inhibitory control and related constructs of attention focusing and effortful control; the instrument is a modified form of the one developed by Rothbart [28,29]. CBCL is a questionnaire administered to assess competencies and behavior problems of children and adolescents [30]; it provides a profile of the child’s social and behavioral functioning relative to children of the same age and sex. The CBCL was administered verbally by a trained research interviewer to ensure uniform administration across sites to all caregivers regardless of literacy level. From the summary scores, we derived the scores for the attention problems scale for analysis in this study.

Statistical Analysis: We computed descriptive statistics, including mean (standard deviation, SD) and proportions, and used ANOVA and chi-square tests to compare demographic characteristics of participants by levels of prenatal cocaine exposure and by diagnosis of ADHD. Logistic regression analyses were conducted using SAS PROC SURVEYLOGISTIC (version 9.4) to calculate adjusted odds ratios of ADHD by demographic characteristics, accounting for clustering of participants by site. The path analysis was constructed with the predictor variables and the mediators of the diagnosis of ADHD at 11 or 14 years. The predictor variable was the prenatal drug exposure categorized as levels of PCE. The independent variables were child’s sex, birth weight, preterm birth, maternal education, caretaker psychopathology, and caretaker–child conflict. The mediator variables were child’s impulsivity and CBCL attention problems at 5 years. In the interest of model parsimony, the path models were trimmed by removing any paths that ended up with *p*-values > 0.1.

Generative artificial intelligence (GenAI) has been used in this paper to search for references and generate some part of text.

## 3. Results

### 3.1. Descriptive, Regression, and Path Analyses

#### 3.1.1. Demographic Characteristics (Table 1)

Of the 1388 children enrolled in the follow-up study, 853 children (62%) had an assessment for ADHD and were included in the analysis. Table 1 shows the characteristics of the participants by levels of PCE. Of these children, 11.1% had high PCE/OD and 23.2% had some PCE/OD. Forty percent were in the PCE−/OD+ group, and 25.4% had neither PCE nor other drug exposure (PCE−/OD−). Of those with high PCE, 48% were male, 81% black, and 40% had birth weight < 2500 g. Of the mothers of children with high PCE, 53% had less than a high school education and had higher scores in measures of psychopathology and in caretaker–child conflict. As to mediators, child impulsivity and attention problems are also noted in this table by levels of PCE.

#### 3.1.2. ADHD Diagnosis by Demographics (Table 2)

In total, 106 (12.4%) met criteria for ADHD, while 747 (87.6%) did not have the diagnosis. Table 2 shows the demographics of the children by ADHD diagnosis at 11 or 14 years of age. Significantly higher proportions of children with ADHD were male and white. Mothers of children with ADHD had higher scores in psychopathology and caretaker–child conflict than those with children with no ADHD. The unadjusted odds ratios (95% CI) for impulsivity and attention problems in those with ADHD were 6.41 (2.83, 14.50) and 7.09 (3.98, 12.64), respectively (*p* < 0.001).

**Table 2 children-12-01570-t002:** DISC ADHD diagnosis at 11 or 14 years by demographic characteristics.

Characteristic	All	DISC Diagnosis at 11/14 Years			
	N	ADHD N (%)	No ADHD N (%)	Unadjusted OR (95% CI)	*p*
Demographics	853	106 (12.4)	747 (87.6)	-	-
Prenatal drug exposure					
High PCE/OD	95	18 (19)	77 (81)	2.94 (1.43, 6.05)	**0.004**
Some PCE/OD	198	29 (15)	169 (85)	2.16 (1.13, 4.10)	**0.019**
PCE−/OD+	343	43 (13)	300 (87)	1.80 (0.99, 3.29)	**0.055**
PCE−/OD−	217	16 (7)	201 (93)	REF	
Gender					
Male	434	69 (16)	365 (84)	1.95 (1.28, 2.98)	**0.002**
Female	419	37 (9)	382 (91)	REF	
Race/Ethnicity					
Black	683	78 (11)	605 (89)	0.53 (0.31, 0.90)	**0.019**
White	107	21 (20)	86 (80)	REF	
Other	63	7 (11)	56 (89)	0.51 (0.20, 1.28)	0.153
Birth weight					
<2500 g	342	47 (14)	295 (86)	1.22 (0.81, 1.84)	0.341
≥2500 g	511	59 (12)	452 (88)	REF	
Maternal age at birth	853	27.8 ± 5.5	27.7 ± 5.9	1.00 (0.97, 1.04)	0.954
Education					
Less than high school	312	43 (14)	269 (86)	1.09 (0.64, 1.86)	0.764
High school	354	39 (11)	315 (89)	0.84 (0.49, 1.45)	0.531
More than high school	187	24 (13)	163 (87)	REF	
Caretaker psychopathology (BSI) (4 months to 5 years)	828	0.7 ± 0.6	0.6 ± 0.5	1.99 (1.37, 2.88)	**<0.001**
Relationship with child: Conflict score at 5 years	602	2.7 ± 0.7	2.4 ± 0.7	1.81 (1.31, 2.50)	**<0.001**
CBQ child impulsivity score at 4 years	658	2.5 ± 0.3	2.3 ± 0.3	6.41 (2.83, 14.50)	**<0.001**
CBCL attention problems at 5 years	61	27(44)	34 (56)	7.09 (3.98,12.64)	**<0.001**
	536	54 (10)	482 (90)	REF	

N (%); mean ± SD; REF = reference category; 4 categories: high PCE with other drug exposure (high PCE/OD), some PCE/OD, no PCE but with exposure to OD (PCE−/OD+), and no PCE and no OD exposure (PCE−/OD−).

#### 3.1.3. ADHD Diagnosis According to Levels of PCE (Figure 1)

Figure 1 shows the percentage of children with ADHD according to levels of PCE in male and female children. The incidence of ADHD was highest in male children who had high PCE/OD at 25%, decreasing to 19% in some PCE/OD, and to 15% in those with no PCE but with other drug exposure. The percentage of male children with PCE−/OD− was 11%, higher than in female children (9%) with high PCE. As shown in Figure 1, the percentage of ADHD among female children was consistently lower than in males, regardless of the level of PCE.

**Figure 1 children-12-01570-f001:**
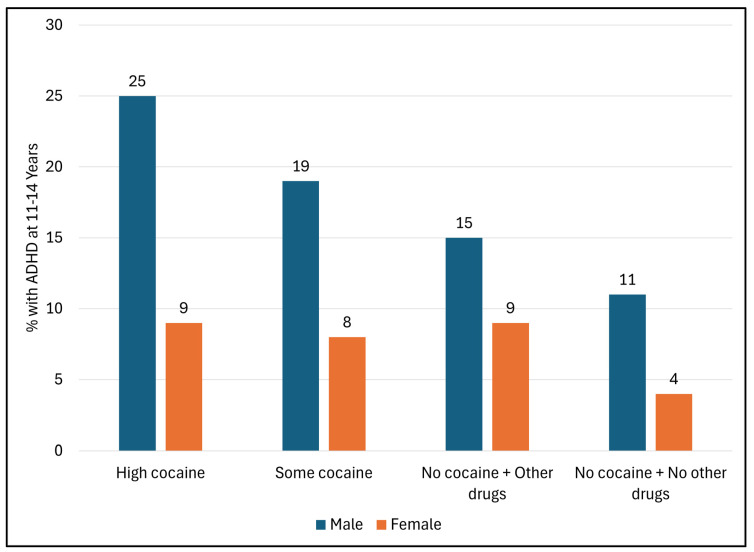
Percentages of participants with ADHD diagnosis by levels of prenatal cocaine exposure in male and female children. Note: Percentages are adjusted for male sex, Black race, birth weight < 2500 g, less than high school education, caretaker BSI, and relationship with child: conflict score.

#### 3.1.4. Logistic Regression Analyses (Table 3)

Table 3 shows the adjusted odds ratios and 95% confidence interval (CI) from the logistic regression models of ADHD diagnosis at 11 or 14 years. High PCE was a significant predictor of ADHD, with an adjusted odds ratio (95% CI) of 2.56 (1.02, 6.41). Other significant predictors were sex, low birth weight, caretaker psychopathology, and caretaker–child conflict. The child’s race tended to be associated with ADHD with odds ratio (95% CI) of 0.58 (0.33, 1.02), *p* = 0.06. Maternal education was not significantly associated with ADHD (*p* = 0.98).

**Table 3 children-12-01570-t003:** Logistic regression models of ADHD diagnosis at 11 or 14 years.

Variable	N (%) with ADHD	Adjusted OR (95%CI)	*p*
Prenatal drug exposure			
High PCE/OD	18 (19)	2.56 (1.02, 6.41)	**0.045**
Some PCE/OD	29 (15)	1.96 (0.90, 4.27)	0.092
PCE−/OD+	43 (13)	1.70 (0.83, 3.48)	0.148
PCE−/OD−	16 (7)	REF	
Sex			
Male	69 (16)	2.38 (1.41, 4.03)	**0.001**
Female	37 (9)	REF	
Race/Ethnicity			
Black	78 (11)	0.58 (0.33, 1.02)	0.060
White/Other	28 (16)	REF	
Birth weight			
<2500 g	47 (14)	1.76 (1.06, 2.91)	**0.029**
≥2500 g	59 (12)	REF	
Education			
Less than high school	43 (14)	1.01 (0.60, 1.70)	0.976
High school or more	63 (12)	REF	
Caretaker BSI (4 months to 5 years)	--	1.74 (1.10, 2.75)	**0.018**
Relationship with child—Conflict score (5 years)	--	1.04 (1.01, 1.07)	**0.008**

Note: REF = reference category. Model also accounts for clustering of children by research center. Diagnoses of inattention, hyperactivity, or combined based on prior year symptoms at 11-year and 14-year visits. Four categories: high PCE with other drug exposure (high PCE/OD), some PCE/OD, no PCE but with exposure to OD (PCE−/OD+), and no PCE and no OD exposure (PCE−/OD−).

#### 3.1.5. Path Analysis (Figure 2)

A path diagram is shown in Figure 2. The model fit well (CFI = 0.998, TLI = 0.996, and RMSEA = 0.005). High PCE was a significant predictor of child impulsivity at age 4 years (*p* = 0.043), which was in turn a significant predictor of ADHD diagnosis at 11/14 years of age (*p* = 0.006). Impulsivity was also significantly associated with attention problems at age 5 years (*p* = 0.017). High PCE (*p* = 0.034) and some PCE (*p* = 0.0001) were significant risks in the path to attention problems, which were associated with ADHD at 11 and/or 14 years (*p* < 0.001). Other significant factors associated with CBCL attention problems included low birth weight (*p* = 0.046), caretaker psychopathology (*p* = 0.024), and caretaker conflict with child (*p* < 0.001). Male sex had a significant direct path to ADHD (*p* < 0.001) and was not mediated by impulsivity or attention problems at 4 and 5 years, respectively.

**Figure 2 children-12-01570-f002:**
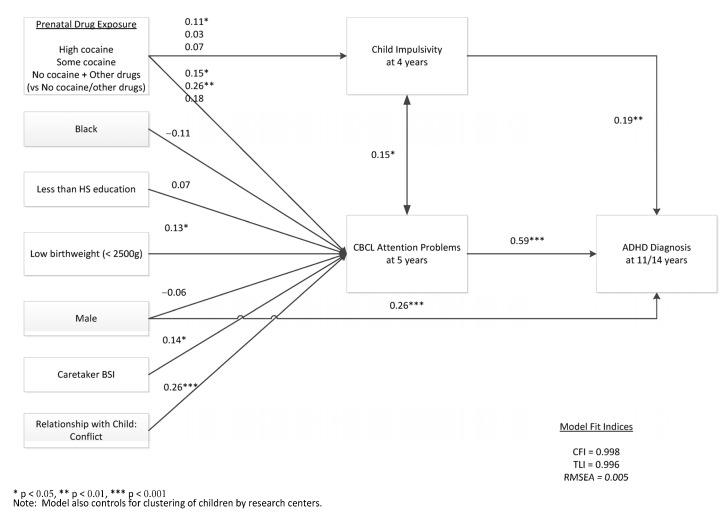
Path diagram of ADHD Diagnosis at 11 or 14 years. The risk factors and mediator variables (impulsivity and attention problems) are shown with the coefficients and corresponding notation of significance (*p* values).

## 4. Discussion

Our study highlights the complex interplay between PCE, neurobehavioral characteristics, and the later development of ADHD. Among the children analyzed, 12.4% met criteria for ADHD by ages 11 or 14 years. Although PCE emerged as an important early risk factor, its effect on ADHD was largely mediated by intermediary behavioral traits rather than a direct association. Consistent with prior research, low birth weight, maternal psychological symptoms, and maternal–child conflict were also linked to ADHD. However, these influences operated through early behavioral markers, impulsivity at age 4 [31] and attention problems at age 5 [32]. Male sex, in contrast, emerged as a significant risk and showed a direct relationship with ADHD, independent of these mediators.

These findings underscore the critical role of early behavioral characteristics as developmental precursors. Impulsivity and attention problems appear to channel environmental and biological risks into clinical outcomes. Path analysis further supports this mediation pathway, showing that early impulsivity and attention problems significantly increased the odds of later ADHD, even after adjusting for demographic and environmental variables.

Our results are consistent with previous studies that have reported associations between PCE/polydrug exposure and attention problems in children [33,34,35]. In our study, although PCE was a strong predictor of both impulsivity and attention difficulties, it was not independently associated with ADHD in the final model. This suggests that its impact may be primarily indirect, operating through early behavioral traits. Our results align with earlier report which identified third-trimester PCE as a predictor of inattention and impulsivity at age seven, preceding the ADHD diagnoses observed at later ages in our study. Mediation analysis further indicated that total behavior problems at age three mediated the effect of third-trimester exposure on later behavioral outcomes [36].

Although causality cannot be inferred from our study, prior research has linked PCE to later ADHD symptoms through disruption of developing monoaminergic systems critical for attention and executive control [37]. Experimental and neuroimaging studies demonstrate altered dopamine transporter expression [38], down-regulation of α_2_-adrenergic autoreceptors [39], abnormalities in frontal–striatal circuits, white-matter integrity, and structural brain maturation [40,41,42]. Collectively, these findings support the notion that PCE produces long-lasting catecholaminergic dysregulation and disruption of executive-function networks.

Maternal-related factors, including psychopathology and parent–child conflict, were also associated with increased ADHD risk, likely through their influence on early child behavior and self-regulation. Caregivers with poorer mental health have been reported to have higher odds of raising a child with severe ADHD, suggestive of the caregiver’s emotional health playing a role in childhood ADHD prevalence [43]. Conversely, a positive maternal–child interaction as reflected by maternal warmth or expressed enthusiasm, interest in, and enjoyment of the child, has been shown to moderate the ADHD symptoms in children, with more pronounced effect on those born with low birth weight [44]. Children with birth weight of less than 2500 g are 3 times more likely to develop ADHD, even after controlling for factors or confounders such as prenatal alcohol and tobacco exposure, parental ADHD, and socioeconomic status [45]; they are also more likely to exhibit ADHD symptoms than their higher birth weight siblings [46].

Interestingly, male sex was directly associated with ADHD but not with earlier attention problems. This suggests a potential sex-specific pathway to diagnosis that may be independent of early observable behavior; this raises important questions about sex differences in symptom presentation, diagnosis, and underlying neurobiology [47]. It is worth mentioning, however, that a growing body of work indicates that females with ADHD are less likely to be recognized in childhood and are diagnosed later than males—often after intervening diagnoses of anxiety/depression—resulting in delayed or missed access to treatment and educational supports [48,49,50].

Overall, our results emphasize the importance of early identification and evaluation for intervention, particularly among children with known prenatal drug exposures or early behavioral problems. The American Academy of Pediatrics recommends that evaluation for ADHD be initiated as early as 4 years of age in children with academic or behavioral difficulties [51]. Yet, in the U.S. data from 2022 indicate that 2.4% of children, aged 3 to 5 years, had an ADHD diagnosis; up to 40.6% received neither medication nor behavioral intervention [6]. Of concern, 65% of these children with ADHD had co-occurring disorders (mental, emotional, behavioral, developmental, language, or autism spectrum disorders). With treatment, however, favorable outcomes were reported as compared to those with no treatment or as improvement or stabilization from pretreatment baseline [52]. Given these findings, screening for ADHD and comorbidities may be warranted even before age 4. Behavior screening is best incorporated with the routine early childhood developmental screening [53,54] to identify at-risk children. Positive behavior screens can then be followed by a validated behavior assessment tool to define specific behavior concerns. Importantly, most tools also assess for co-occurring behavioral issues, allowing comprehensive evaluation at early ages of 2 to 3 years [53,55,56,57,58]. Early detection of behavioral problems enables timely behavioral intervention such as Parent Training Intervention and Education programs, which have been reported to result in robust improvement in child outcomes, reduced ADHD symptoms and disruptive behaviors, and improve parenting skills, with parent-reported positive sense of competence and parent–child relationship [59,60,61,62]. During early intervention, the risk for emerging health issues associated with ADHD can be monitored and health promotion strategies can be implemented [10].

The limitations of this study include the use of secondary data from the MLS, where ADHD diagnoses were not based on DSM-5 criteria or direct clinical assessment and/or observations in different settings. We used both maternal interviews and meconium analyses to include the participants in the study; this approach yielded a higher prevalence rate of drug exposure identification and allowed us to categorize the levels of exposure into heavy vs. light exposure [63]. We did not analyze for behavioral comorbidities, environmental risks, foster care and other home placement, endocrine disruptors, pharmacologic interventions, genetic, and other potential influences on ADHD risk. Despite these limitations, our path analysis clarifies how perinatal risks and caretaker relationships contribute to ADHD development through early behavioral mediators.

## 5. Conclusions

Our study demonstrates that PCE is an important early risk factor for ADHD, but its effects are largely mediated through early behavioral traits—specifically impulsivity at age 4 and attention problems at age 5. Male sex emerged as a direct, independent risk factor. Caregiver psychopathology, maternal–child conflict, and low birth weight also contributed indirectly through their influence on early behavior.

For clinicians, our findings underscore the importance of early behavioral screening and intervention in children at risk, particularly those with prenatal substance exposure or perinatal risk factors. Identifying impulsivity and attention difficulties in the preschool years may provide opportunities to intervene before symptoms progress to full ADHD. Future research should evaluate the effectiveness of targeted early interventions and further explore sex-specific pathways and biological mechanisms underlying ADHD risk.

We hope our findings will help guide healthcare policymakers in developing strategies to support early identification and intervention. Parental training and early behavioral support could mitigate long-term impairment, enhance children’s developmental, educational, and social functioning, thereby benefiting families and educators. Nevertheless, the implementation of such strategies would require investment and resource allocation that warrant careful evaluation.

## Figures and Tables

**Table 1 children-12-01570-t001:** Demographic characteristics by prenatal drug exposure.

Characteristic	All N = 853	High PCE/OD N= 95 (11.1)	Some PCE/OD N = 198 (23.2)	PCE−/OD+ N = 343 (40.2)	PCE−/OD− N = 217 (25.4)
	N (%)	N (%)	N (%)	N (%)	N (%)
Demographics					
Gender					
Male	434 (51)	46 (48)	101 (51)	175 (51)	112 (52)
Female	419 (49)	49 (52)	97 (49)	168 (49)	105 (48)
Race/Ethnicity					
Black	683 (80)	77 (81)	166 (84)	257 (75)	183 (84)
White	107 (13)	12 (13)	18 (9)	60 (17)	17 (8)
Other	63 (7)	6 (6)	14 (7)	26 (8)	17 (8)
Birth weight					
<2500 g	342 (40)	38 (40)	83 (42)	121 (35)	100 (46)
≥2500 g	511 (60)	57 (60)	115 (58)	222 (65)	117 (54)
Less than high school education	312 (37)	50 (53)	91 (46)	117 (34)	54 (25)
Caretaker psychopathology (BSI) (4 months to 5 years), mean (SD)	0.6 (0.5)	0.7 (0.5)	0.6 (0.5)	0.6 (0.5)	0.5 (0.5)
Relationship with child: Conflict score at 5 years), mean (SD)	28.9 (8.7)	31.1 (8.8)	30.5 (8.8)	28.9 (8.1)	26.9 (9.1)
Mediators					
CBQ child impulsivity score at 4 years, mean (SD)	2.3 (0.3)	2.4 (0.3)	2.4 (0.3)	2.4 (0.3)	2.3 (0.3)
CBCL Attention problems at 5 years	61 (10)	8 (12)	25 (19)	22 (9)	6 (4)

Four categories: high PCE with other drug exposure (high PCE/OD), some PCE/OD, no PCE but with exposure to OD (PCE−/OD+), and no PCE and no OD exposure (PCE−/OD−); N (%); mean ± SD.

## Data Availability

The datasets presented in this article are not readily available as these are only a secondary analysis, part of the Maternal Lifestyle Study (MLS), a multicenter longitudinal study. Requests to access the datasets should be directed to H.B. The MLS data of four US sites (1993–2011) can be requested from the National Addiction & HIV Archive Program at https://www.icpsr.umich.edu/web/NAHDAP/studies/34312 (accessed on 31 October 2025).

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
