# Peer review of "Prenatal Cocaine Exposure, Perinatal Risks, and Mediators to Preadolescent Attention Deficit Hyperactivity Disorder (ADHD)"

_children, 2025, doi:10.3390/children12111570_

Round 1
Reviewer 1 Report
Comments and Suggestions for Authors
The aim of the study is to determine: 1. the association between high prenatal cocaine exposure, sex, race, maternal age, maternal education, and ADHD in children; 2. whether the child’s behavior of impulsivity and inattention would mediate the relationship between high prenatal cocaine exposure and ADHD. The study is a secondary analysis of data from the Maternal Lifestyle Study (MLS). The participants of the study were children with prenatal cocaine and/or opiate exposure and their comparison cohort. Authors revealed that prenatal cocaine exposure is early risk factor for ADHD diagnosis at 11/14 years and its effects are mediated through impulsivity at age 4 years and attention problems at age 5 years. Caregiver psychopathology, maternal-child conflict, low birth weight contributed ADHD at 11/14 years indirectly through their influence on child’s behavior at 4-5 years. Male sex is direct risk factor of ADHD at 11/14 years.
Strengths of the paper:
The paper is well written; the topic of the study is relevant, since it is associated with finding of early ADHD risk factors and markers and possibility of early intervention in children with perinatal risk factors.
Questions and remarks:
- The section Materials and Methods does not clearly describe the number of participants – children with prenatal cocaine and/or opiate exposure and their comparison cohort.
- How were children aged 4-5 years to 11/14 treated (medications, other therapy)? Could treatment have contributed to ADHD at 11/14 years?
- Authors noted: “Exposed children were those whose mothers admitted to any cocaine or opiate use during pregnancy or whose children’s meconium showed cocaine or opiate metabolites.” Are interviews with mothers a reliable source of information about how much and what drugs they used during pregnancy?
- For some children, parents gave consent for study, for some children - legal guardian. If these were foster families, at what age did the child enter the foster family? What was the child’s background before entering the foster family?
- The questions that arise when reading the paper could be removed by the section Limitation.
Technical notes
- Figure 1: There is too much empty space in this figure. Why is the vertical axis 100%? Is it possible to reduce the scale to 30 or 40%?
- There are many abbreviations in the text. Please check their explanations. Why is the abbreviation PCE explained again in the section Conclusion? Decipher abbreviation AOR in table 3.
Author Response
We are grateful to Reviewer #1 for the careful reading of our manuscript and the constructive suggestions. We appreciate the recognition of the study’s relevance and quality of writing. In response to the reviewer’s valuable comments, we have clarified the Methods section, added additional details to the Discussion, and made technical adjustments to the figures and tables as outlined below.
Comment 1: The section Materials and Methods does not clearly describe the number of participants – children with prenatal cocaine and/or opiate exposure and their comparison cohort.
Response 1: We have added the number of participants for the exposure and comparison groups in the materials and methods (page 3 line 120, page 4 line 121-122) and into the column headers for Table 1.
Comment 2: How were children aged 4-5 years to 11/14 treated (medications, other therapy)? Could treatment have contributed to ADHD at 11/14 years?
Response 2: The MLS was designed to better understand the neurodevelopmental course of children with prenatal cocaine and or opioid exposure, and thus measures of family, environmental changes, and different domains in development were prospectively evaluated. The medications or any specific treatment for ADHD were not collected.
We have added as limitation” “We also lacked detailed data on behavioral, environmental including details of foster parenting, endocrine disruptors, or pharmacologic interventions for ADHD for these children, as well as genetic and other potential influences on ADHD risk.” Please see discussion section, page 12 line 368-371.
Comment 3: Authors noted: “Exposed children were those whose mothers admitted to any cocaine or opiate use during pregnancy or whose children’s meconium showed cocaine or opiate metabolites.” Are interviews with mothers a reliable source of information about how much and what drugs they used during pregnancy?
Response 3: We have added as limitation “We used both maternal interviews and meconium analyses to include the participants in the study. MLS reported using both approaches yielded a higher prevalence rate of drug exposure. Interviews are reliable and informative, particularly when combined with biologic measures where feasible and can detect cocaine exposures if missed with the other approach and allowed us to categorize the levels of exposure into heavy vs light exposure” in discussion page 12, line 363-368.
Comment 4: For some children, parents gave consent for study, for some children - legal guardian. If these were foster families, at what age did the child enter the foster family? What was the child’s background before entering the foster family?
Response 4: We have added “We also lacked detailed data on behavioral, environmental including details of foster parenting, endocrine disruptors, or pharmacologic interventions for ADHD for these children, as well as genetic and other potential influences on ADHD risk” as limitations in the discussion section, page 12 line 368-371.
Foster issue is quite complex, many children had multiple foster parents already in the earlier years.
5. The questions that arise when reading the paper could be removed by the section Limitation.
Technical notes
1. Figure 1: There is too much empty space in this figure. Why is the vertical axis 100%? Is it possible to reduce the scale to 30 or 40%?
We have revised Figure 1, so the vertical axis ranges from 0% to 30%
2. There are many abbreviations in the text. Please check their explanations. Why is the abbreviation PCE explained again in the section Conclusion? Explanation for PCE abbreviation is now removed from the Conclusion Section. Decipher abbreviation AOR in table 3.
We have revised Table 3 to change AOR to “Adjusted OR.”
Reviewer 2 Report
Comments and Suggestions for Authors
This is a very interesting work, but there are several aspects that must be considered to assess the novelty of the information they provide: The age of the data (1993-1995 ~ 2007-2009) and the existence of works that deal with the objectives of analysis of this research with the same sample. Particularly the work of Bada et al. (2007) [https://doi.org/10.1542/peds.2006-1404].
The work of Richardson et al. (2011) [https://doi.org/10.1016/j.ntt.2010.06.003] with part of the sample used for this work, found: "Cocaine use in the third trimester was also a significant predictor of increased activity scores (Routh and SNAP), as well as increased inattention, impulsivity, and peer problems (SNAP), with effect sizes ranging from 1.7 to 4.7 points (Table 6)."
They also performed moderation and mediation analyses related to sociodemographic factors directly related to hypothesis 2 of this study.
This text is not very friendly to the reader. Regardless of the doubts it generates in me. In view of the characteristics of the sample, I have doubts about the object of the analyses that have been carried out. I have doubts about the techniques used and how they have been interpreted.
The headings of the Results section are 3.1. (EMPTY). 3.1.1. Table 1; 3.1.2. Table 2; 3.1.3. Figure 1; 3.1.4. Table 3; 3.1.5. Figure 2. I think it would be more appropriate for them to refer, for example, to research hypotheses.
This is an aspect that has been an added difficulty in the review. First, the justification of the analysis techniques in relation to the analysis hypotheses, and second, that although the Odds values of children diagnosed with ADHD in the group with Prenatal Exposure to Cocaine are presented, a comparison is not made with the Odds experducable in a control group (I have not checked whether these data are included in the National Cocaine database). Addiction & HIV Data Archive Program).
Figure 2 is an essential part of the study. On the one hand, the sample may be biased, making it difficult to assume the results with confidence. The coefficients, regardless of whether the system tells us that they are significant, are relatively small. I think that the use of confirmatory analysis techniques can be valued.
A high relationship (B=0.59) was found between Child Behavior Checklist scores at age 5 and ADHD diagnosis between ages 11 and 14. I think it was to be expected. It would possibly have been reasonable to verify the diagnosis of ADHD in cases with evidence of risk after administering the Child Behavior Checklist instrument. The strange thing is that this relationship had not occurred.
This is a relevant aspect in research design; different variables must measure different things. If there is overlap between what the variables measure, the explanatory model (the regression equation) will not be well weighted.
In the concussions section, I think that considering that the male sex is a direct and independent risk factor, as well as the consideration that there is an evolution toward complete ADHD can be questionable.
References:
Consider the possibility of including other related works that were carried out using the same sample, for example:
Nygaard E, Slinning K, Moe V, Walhovd KB (2016) Behavior and Attention Problems in Eight-Year-Old Children with Prenatal Opiate and Poly-Substance Exposure: A Longitudinal Study. PLoS ONE 11(6): e0158054. https://doi.org/10.1371/journal.pone.0158054
https://doi.org/10.1001/jamapsychiatry.2013.1949
https://doi.org/10.1001/jamapediatrics.2013.550
https://doi.org/10.1097/DBP.0b013e3182560cd9
https://doi.org/10.1016/j.ntt.2010.06.005
https://doi.org/10.1111/jspn.12358
https://doi.org/10.1097/01.DBP.0000285681.40235.4f
Ackerman, L.E. (2011). Moving Towards a Differential Diagnosis: ADHD versus Depression in Young Children. https://www.proquest.com/openview/1353cd14b71380603b982c83f726ebca/1?pq-origsite=gscholar&cbl=18750

Author Response
We appreciate Reviewer #2’s thoughtful review and constructive feedback concerning the novelty of our work, the use of the MLS dataset, and the clarity of our analyses. In response, we have emphasized how our study extends prior MLS investigations by focusing on ADHD diagnosis at 11/14 years, clarified the analytic approach, refined the structure of the Results section, and incorporated additional references to relevant studies. Detailed responses to each comment are outlined below.
Comment 1: This is a very interesting work, but there are several aspects that must be considered to assess the novelty of the information they provide: The age of the data (1993-1995 ~ 2007-2009) and the existence of works that deal with the objectives of analysis of this research with the same sample. Particularly the work of Bada et al. (2007) [https://doi.org/10.1542/peds.2006-1404].
Response 1: The work noted in the above comment had the objective of examining the trajectories of externalizing, internalizing, and total behavior problems after prenatal cocaine/opiate exposure till age 7 years. These behavior problems are broad band measures from the CBCL administration. Although risk factors were also addressed for these behavior problems, the outcome ADHD diagnosis was not assessed until the children were 11 and 14 years old.
Comment 2: The work of Richardson et al. (2011) [https://doi.org/10.1016/j.ntt.2010.06.003] with part of the sample used for this work, found: "Cocaine use in the third trimester was also a significant predictor of increased activity scores (Routh and SNAP), as well as increased inattention, impulsivity, and peer problems (SNAP), with effect sizes ranging from 1.7 to 4.7 points (Table 6).
Response 2: We thank the reviewer for the information. Our study categorized PCE by levels of exposure in the first trimester and found significant effect on outcome ADHD.
We referenced Richardson’s study “Our results align with earlier report which identified third-trimester PCE as a predictor of inattention and impulsivity at age seven, preceding the ADHD diagnoses observed at later ages in our study. Mediation analysis further indicated that total behavior problems at age three mediated the effect of third-trimester exposure on later behavioral outcomes” in the discussion, page 11, line 298-302.
Comment 3: They also performed moderation and mediation analyses related to sociodemographic factors directly related to hypothesis 2 of this study.
This text is not very friendly to the reader. Regardless of the doubts it generates in me. In view of the characteristics of the sample, I have doubts about the object of the analyses that have been carried out. I have doubts about the techniques used and how they have been interpreted.
The headings of the Results section are 3.1. (EMPTY). 3.1.1. Table 1; 3.1.2. Table 2; 3.1.3. Figure 1; 3.1.4. Table 3; 3.1.5. Figure 2. I think it would be more appropriate for them to refer, for example, to research hypotheses.
Response 3: We have added the subheading title to each section of the results. The font and format were fitted to the Journal’s template.
Comment 4: This is an aspect that has been an added difficulty in the review. First, the justification of the analysis techniques in relation to the analysis hypotheses, and second, that although the Odds values of children diagnosed with ADHD in the group with Prenatal Exposure to Cocaine are presented, a comparison is not made with the Odds experducable in a control group (I have not checked whether these data are included in the National Cocaine database). Addiction & HIV Data Archive Program).
Response 4: The analyses included both children with cocaine exposure and a comparison group. As shown in Table 1, the children with cocaine exposure are split into two groups (High cocaine and Some cocaine) and the comparison (unexposed) children are split into two groups (No cocaine + Other drugs and No cocaine + No other drugs). We have included odds ratios for these groups of exposed and unexposed children in Tables 2 and 3.]
The MLS data of four US sites (1993-2011) can be requested from the National Addiction & HIV Archive Program at https://www.icpsr.umich.edu/web/NAHDAP/studies/34312
(now included in data availability section)
Comment 5: Figure 2 is an essential part of the study. On the one hand, the sample may be biased, making it difficult to assume the results with confidence. The coefficients, regardless of whether the system tells us that they are significant, are relatively small. I think that the use of confirmatory analysis techniques can be valued.
A high relationship (B=0.59) was found between Child Behavior Checklist scores at age 5 and ADHD diagnosis between ages 11 and 14. I think it was to be expected. It would possibly have been to verify the diagnosis of ADHD in cases with evidence of risk after administering the Child Behavior Checklist instrument. The strange thing is that this relationship had not occurred.
This is a relevant aspect in research design; different variables must measure different things. If there is overlap between what the variables measure, the explanatory model (the regression equation) will not be well weighted.
In the concussions section, I think that considering that the male sex is a direct and independent risk factor, as well as the consideration that there is an evolution toward complete ADHD can be questionable.
Comment: we thank the reviewer for the comments.
References:
Comment: Consider the possibility of including other related works that were carried out using the same sample, for example:
Nygaard E, Slinning K, Moe V, Walhovd KB (2016) Behavior and Attention Problems in Eight-Year-Old Children with Prenatal Opiate and Poly-Substance Exposure: A Longitudinal Study. PLoS ONE 11(6): e0158054. https://doi.org/10.1371/journal.pone.0158054
Comment: We have cited this article in the discussion “Our results are consistent with previous studies that have reported associations between PCE/ polydrug exposure and attention problems in children”, page 11, line 294-295
Reviewer 3 Report
Comments and Suggestions for Authors
Dear Authors, This is a very important area of research and it has been a great pleasure to read about this study. The comments below are offered to enhance the manuscript, increase reader interest and citation.
- The reason for undertaking this study is not clearly stated in the introduction. The final paragraph states what was measured, but there was no comment on how this information could be used to help the young people with the symptoms of ADHD, nor their parents, nor their teachers: "We previously reported....For the current study, we focused on maternal and child variables that may be associated with ADHD in adolescence....We determined....". The abstract clearly states that earlier screening for children with PCE for ADHD is practical implication of the study, but the introduction does not explain pre-school screening and then earlier diagnosis could be used to help the young people, their parents or teachers. Please consider expanding the introduction to make these points clear for people not already very familiar with ADHD, presentations, challenges, co-morbidity, interventions and possible positive outcomes of effective support.
- Please replace the word subject with the word participant throughout the manuscript when referring to people.
- Please add some information about the possible neurological mechanism(s) by which PCE could increase the risk of ADHD. Please also comment on other substances and explain why these were not individually analysed.
- In the discussion, could the short paragraph starting: "Interestingly, male sex was directly associated with ADHD but not with earlier attention problems" be expanded as there is a growing body of work on the late/under-diagnosis of girls with ADHD resulting in later or no support for them.
- In the discussion please consider adding to the comments about the effectiveness of early intervention: "Early detection of behavioral problems enables timely behavioral intervention such as Parent Training Intervention and Education programs, which have been reported to result in robust improvement in child outcomes (50-52).", by adding some information about the parents' feedback on taking part in such an intervention and the effect on their child.
- Line 326, Please consider replacing the word "Alarmingly..." with a less emotive word or phrase, while still raising this is a serious concern.
- The conclusion advises that pre-school screening and intervention could potentially reduce the symptoms of ADHD for the children. Following this argument it is logical that this could improve the children's development, education and social interactions, which would be of benefit for the children, their parents and their teachers. However, this would also incur a cost. Please discuss which organization(s) should organise the early screening and which organization(s) or individuals should pay for this.
Author Response
Comment: Dear Authors, This is a very important area of research and it has been a great pleasure to read about this study. The comments below are offered to enhance the manuscript, increase reader interest and citation.
We sincerely thank Reviewer #3 for the positive and encouraging comments on our manuscript and for recognizing the importance of this area of research. We greatly appreciate the thoughtful suggestions provided to enhance the clarity, depth, and practical relevance of the paper. We have carefully addressed each point by expanding the introduction, enriching the discussion with additional mechanistic and contextual details, refining terminology, and elaborating on the implications for early intervention and policy.
Comment 1: The reason for undertaking this study is not clearly stated in the introduction. The final paragraph states what was measured, but there was no comment on how this information could be used to help the young people with the symptoms of ADHD, nor their parents, nor their teachers: "We previously reported....For the current study, we focused on maternal and child variables that may be associated with ADHD in adolescence....We determined....". The abstract clearly states that earlier screening for children with PCE for ADHD is practical implication of the study, but the introduction does not explain pre-school screening and then earlier diagnosis could be used to help the young people, their parents or teachers. Please consider expanding the introduction to make these points clear for people not already very familiar with ADHD, presentations, challenges, co-morbidity, interventions and possible positive outcomes of effective support.
Response 1: We have modified and added “Our prior findings from school-aged assessments using the Child Behavior Checklist (CBCL) demonstrated that PCE was associated with elevated attention problem scores reported by both parents or caregivers and teachers, although an ADHD diagnosis had not been established in that cohort. Therefore, this study aimed to further determine whether preschool-age screening for behavioral problems, along with maternal and child variables, is associated with a subsequent diagnosis of ADHD during adolescence. Identification of such risk factors and comorbidities may facilitate early intervention and contribute to improved long-term outcomes” in the introduction, page 3, line 81-88.
Comment 2: Please replace the word subject with the word participant throughout the manuscript when referring to people.
Response 2: We have replaced the word subject with participant.
Comment 3: Please add some information about the possible neurological mechanism(s) by which PCE could increase the risk of ADHD. Please also comment on other substances and explain why these were not individually analysed.
Response 3: We have expanded the possible neurological mechanisms “Although the causation may not be concluded from our study, previous studies reported association between PCE and later ADHD symptoms that may be explained by cocaine’s disruption of developing monoaminergic systems, particularly dopaminergic and noradrenergic pathways that regulate attention and executive control. Experimental and neuroimaging studies demonstrate that prenatal exposure alters dopamine transporter expression, down-regulates αâ‚‚-adrenergic autoreceptors in the locus coeruleus, and produces long-lasting hyper-responsivity of the catecholaminergic system, leading to deficits in arousal and attention regulation. Structural and functional MRI findings further show abnormalities in frontal–striatal circuits and white-matter integrity among exposed adolescents, consistent with impaired executive networks. Together, these neurobiological alterations provide a plausible mechanistic link between prenatal cocaine exposure and increased risk of ADHD in later childhood and adolescence” in the discussion, page 11, line 303-314.
The effect of substances was not individually analyzed because multidrug use was common among cocaine users and comparison group in the study.
Comment 4: In the discussion, could the short paragraph starting: "Interestingly, male sex was directly associated with ADHD but not with earlier attention problems" be expanded as there is a growing body of work on the late/under-diagnosis of girls with ADHD resulting in later or no support for them.
Response 4: We have added “It is worth mentioning however, that a growing body of work indicates that females with ADHD are less likely to be recognized in childhood and are diagnosed later than males—often after intervening diagnoses of anxiety/depression—resulting in delayed or missed access to treatment and educational supports”. In the discussion, page 11 line 330 to page 12 line 334
Comment 5: In the discussion please consider adding to the comments about the effectiveness of early intervention: "Early detection of behavioral problems enables timely behavioral intervention such as Parent Training Intervention and Education programs, which have been reported to result in robust improvement in child outcomes (50-52).", by adding some information about the parents' feedback on taking part in such an intervention and the effect on their child.
Response 5: We have modified the discussion, “Early detection of behavioral problems enables timely behavioral intervention such as Parent Training Intervention and Education programs, which have been reported to result in robust improvement in child outcomes, reduced ADHD symptoms and disruptive behaviors and improve parenting skills; parent-reported positive sense of competence and parent-child relationship” the discussion page 12, line 350-354.
Comment 6: Line 326, Please consider replacing the word "Alarmingly..." with a less emotive word or phrase, while still raising this is a serious concern.
Response 6: We have replaced with the word “Of concern, now page 12. Line 341
Comment 7: The conclusion advises that pre-school screening and intervention could potentially reduce the symptoms of ADHD for the children. Following this argument it is logical that this could improve the children's development, education and social interactions, which would be of benefit for the children, their parents and their teachers. However, this would also incur a cost. Please discuss which organization(s) should organise the early screening and which organization(s) or individuals should pay for this.
Response 7: We have added “We hope our findings will help guide healthcare policymakers in developing strategies to support early identification and intervention. Parental training and early behavioral support could mitigate long-term impairment, enhance children’s developmental, educational, and social functioning, thereby benefiting families and educators. Nevertheless, the implementation of such strategies would require investment and resource allocation that warrant careful evaluation” in the discussion, page 13, line 385-391.
Round 2
Reviewer 2 Report
Comments and Suggestions for Authors
I believe the authors have made a significant effort to modify the content and form of the submitted work.
The authors have made a significant effort to improve the article.
I understand that they have tried to adhere, as far as possible, to the review guidelines.
They have made modifications to both the text and the references.
I believe it is relevant that the authors have included the possibility of underdiagnosis in girls.
In the previous review, we stated: “In the section on concussions, I believe that considering male sex as a direct and independent risk factor, as well as the possibility of progression to full-blown ADHD, may be questionable.”
The inclusion of the following text is appropriate:
“that a growing number of studies indicate that girls with ADHD are less likely to be recognized in childhood and receive a later diagnosis than boys, often after intermediate diagnoses of anxiety or depression, resulting in late or no access to treatment and educational support [50-52].”